# Assessment of the Impact of a Meningitis/Encephalitis Panel on Hospital Length of Stay: A Systematic Review and Meta-Analysis

**DOI:** 10.3390/antibiotics11081028

**Published:** 2022-07-30

**Authors:** Kyle D. Hueth, Philippe Thompson-Leduc, Todor I. Totev, Katherine Milbers, Tristan T. Timbrook, Noam Kirson, Rodrigo Hasbun

**Affiliations:** 1BioFire Diagnostics, LLC, Salt Lake City, UT 84108, USA; kyle.hueth@biomerieux.com (K.D.H.); tristan.timbrook@biomerieux.com (T.T.T.); 2Analysis Group, Inc., Montreal, QC H3B 0G7, Canada; katherine.milbers@analysisgroup.com; 3Analysis Group, Inc., Boston, MA 02199, USA; todor.totev@analysisgroup.com (T.I.T.); noam.kirson@analysisgroup.com (N.K.); 4McGovern Medical School, University of Texas Health Sciences Center at Houston, Houston, TX 77030, USA; rodrigo.hasbun@uth.tmc.edu

**Keywords:** diagnostic techniques, neurological, encephalitis, meningitis, patient care, polymerase chain reaction

## Abstract

Meningitis and encephalitis are central nervous system infections with considerable morbidity and mortality. The BioFire^®^ FilmArray^®^ Meningitis/Encephalitis Panel (multiplex ME panel) can identify pathogens rapidly potentially aiding in targeted therapy and curtail antimicrobial exposure. This systematic review and meta-analysis synthesized the literature on the association between the multiplex ME panel and length of hospital stay (LOS), length of acyclovir therapy, and days with antibiotics. MEDLINE and EMBASE were searched. Only studies presenting novel data were retained. Random-effects meta-analyses were performed to assess the impact of the multiplex ME panel on outcomes. Of 169 retrieved publications, 13 met the criteria for inclusion. Patients tested with the multiplex ME panel had a reduction in the average LOS (mean difference [MD] [95% CI]: −1.20 days [−1.96, −0.44], n = 11 studies). Use of the multiplex ME panel was also associated with a reduction in the length of acyclovir therapy (MD [95% CI]: −1.14 days [−1.78, −0.50], n = 7 studies) and a nonsignificant reduction in the average number of days with antibiotics (MD [95% CI]: −1.01 days [−2.39, 0.37], n = 6 studies). The rapidity of pathogen identification contributes to an overall reduced LOS, reductions in the duration of empiric antiviral utilization, and a nonsignificant reduction in antibiotic therapy.

## 1. Introduction

Central nervous system (CNS) infections such as meningitis and encephalitis can be caused by a variety of organisms including viruses, bacteria, or fungi, and can have outcomes that can range from self-limiting to neurological morbidity and death [1]. The causative agents for both diseases vary according to patient characteristics such as age, immunosuppression, or travel history, with some overlapping and other distinct suspect pathogens for neonates, children, and older adults [2,3,4]. Overall, incident cases of meningitis increased globally between 1990 and 2016, from 2.50 million cases to 2.82 million cases [5]. While deaths due to meningitis have decreased overall, it still accounted for nearly 0.6% of all-age deaths and 3% of deaths in children younger than 5 years as of 2016, with the burden of disease carried mainly by countries with lower socioeconomic status [5]. Encephalitis also remains a significant health concern; research in the United States reported that encephalitis accounts for approximately 20,000 hospitalizations per year, 5.8% of which had a fatal outcome [6].

In community-acquired bacterial meningitis, empiric antimicrobial therapy based on patient characteristics such as age, followed by adjustment to specific antibiotic therapy after the causative organism is identified, is recommended [3]. Similarly, for encephalitis it is recommended that efforts be made to identify the causative organism to guide therapy and to better understand prognosis [4]. Rapid identification of causal organisms is therefore important both from a clinical perspective to prevent neurological sequalae and death, and from an antimicrobial stewardship perspective to prevent unnecessary antibiotic or antiviral exposure. Therefore, rapid diagnostic testing has emerged as a potential avenue to facilitate rapid pathogen identification.

The BioFire^®^ FilmArray^®^ Meningitis/Encephalitis Panel (multiplex ME panel), a comprehensive multiplex polymerase chain reaction (PCR) panel for meningitis and encephalitis (BioFire Diagnostics, LLC, Salt Lake City, UT, USA), detects the 14 most common viral, bacterial, and fungal organisms that cause CNS infections. A 2019 systematic review and meta-analysis reported a sensitivity of 90% (95% CI 86−93%) and a specificity of 97% (95% CI 94−99%) for the multiplex ME panel at identifying the causative pathogen in patients with suspected meningitis/encephalitis [1]. Additionally, a 2022 systematic review found that the multiplex ME panel had an estimated sensitivity of 89.5% (95% CI 81.1−94.4) and specificity of 97.4% (95% CI 94.0−98.9) for all bacteria, though heterogeneity was observed in the panel’s sensitivities for certain pathogens [7]. The multiplex ME panel has also been reported to yield rapid results compared to standard care; turnaround time averaged 2.2 to 6.2 h, compared with 24 h in control groups [8]. 

Following its clearance by the U.S. Food and Drug Administration (FDA) in October 2015, several individual studies have examined the potential impact of the multiplex ME panel on hospital outcomes and treatment patterns. However, to the best of our knowledge, there is no quantitative synthesis of the potential benefit of the multiplex ME panel on hospital length of stay. Furthermore, with the known differences in suspected pathogens, signs, and symptoms between pediatric and adult populations, as well as the potential differences in treatment approaches, patient subgroups merit consideration. Therefore, this systematic review and meta-analysis sought to review and synthesize the current literature describing the impact of the multiplex ME panel on hospital length of stay, days with antibiotics and length of acyclovir treatment. The analysis also sought to explore these associations separately in pediatric populations only.

### Objective

To review and synthesize the current literature describing the association between the use of the multiplex ME panel and length of hospital stay, days with antibiotic therapy, and length of acyclovir treatment. In addition, outcomes were reported among studies that included pediatric patients exclusively.

## 2. Materials and Methods

This systematic review and meta-analysis was performed according to the guidance in the Cochrane Handbook and reported following the Preferred Reporting Items for Systematic Reviews and Meta-Analysis (PRISMA 2020) statement [9,10] (see Appendix A).

### 2.1. Electronic Search and Inclusion Criteria

An electronic literature search was performed on 27 November 2020, using the OvidSP interface. The MEDLINE (including MEDLINE In Process) and EMBASE databases were searched for English-language studies published on or after 2015. A combination of terms to identify the multiplex ME panel, meningitis and/or encephalitis, and length of stay were used. Details on the search strategy may be found in Appendix A.

Publications were retained if they met the following criteria: (1) used the BioFire FilmArray multiplex ME panel to determine the etiology of suspected CNS infections (irrespective of whether the patients were positive by the Panel or culture), (2) reported on patients’ length of hospital stay (i.e., the primary outcome), (3) study compared length of stay of patients tested with the multiplex ME panel to another cohort of patients (i.e., comparative design). Only empirical studies presenting novel data from cohorts of patients were retained (i.e., exclusion of case reports, literature reviews, notes, editorials, etc.)

Two reviewers (K.D.H. and T.I.T. or P.T.-L.) independently reviewed titles and abstracts for assessment of eligibility based on the inclusion criteria listed above. Discrepancies were resolved through discussion. Upon inclusion of publications based on titles and abstracts, a second round of inclusion was performed by the same review team using full-length texts. Then again, discrepancies in the inclusion decisions were resolved through discussion.

### 2.2. Data Extraction and Meta-Analysis

Data extraction of the retained publications was performed by two independent reviewers in a standardized Excel grid. Details about the study design, patient demographic and clinical characteristics and study outcomes were retrieved from the texts. Authors of the original publications were not contacted for this review. While the search strategy was built for the identification of publications reporting the primary outcome (i.e., difference in hospital length of stay), many studies also reported outcomes relevant to the study question, including the length of treatment with antibiotics and acyclovir. Therefore, all three endpoints were meta-analyzed.

Study data were standardized into a format suitable for meta-analysis. For example, when the mean number of hospital days was not directly available from the publication, the mean and standard deviation were approximated using the median and inter-quartile range [11]. 

Random-effects meta-analyses were conducted to determine the impact of the multiplex ME panel on (1) length of hospital stay, (2) length of acyclovir therapy, and (3) days with antibiotics. Outcomes were shown as mean differences (MD), representing the difference in means between cohorts of patients tested with the multiplex ME panel and cohorts of patients testing using standard of care. Analyses were performed using Stata 16 (StataCorp. 2019. Stata Statistical Software: Release 16. College Station, TX, USA: StataCorp LLC).

### 2.3. Stratification by Age

To evaluate whether outcomes differed among pediatric patients, a stratification of the meta-analysis was performed among studies that included pediatric populations only. 

### 2.4. Quality Assessment

Quality assessments were completed by two authors (T.T.T. and P.T.-L.) using the Newcastle-Ottawa Scale for observational studies and the Cochrane Risk of Bias 2 (ROB 2) for randomized trials [12,13]. Differences in quality assessments between authors were resolved through consensus. The Newcastle-Ottawa Scale evaluates for the selection of patients, comparability of patients, and assessment of outcomes. ROB 2 appraises five domains of bias in randomized trials including bias arising from randomization process, and selection of the reported result. Sources of heterogeneity between publications were assessed qualitatively, notably by reporting the study design, date range of data collection, and the mean age of the patients included in each study. Statistical heterogeneity was reported using the I^2^ statistic.

### 2.5. Ethics Compliance

This article is based on previously conducted studies and does not contain any new studies with human participants or animals performed by any of the authors.

## 3. Results

### 3.1. Study Selection and Characteristics

A total of 169 articles were identified by the electronic literature search. Of these, 12 were retained for data extraction (Figure 1). One study was subsequently manually added as it reported novel information on the outcomes of interest in a format suitable for inclusion [14]. While all studies reported the impact of the multiplex ME panel on hospital length of stay, two studies were excluded from the meta-analyses as they did not report enough information to calculate the mean reduction in hospital length of stay [15,16]. Overall, eleven studies reported at least one outcome of interest for inclusion in the meta-analysis [14,17,18,19,20,21,22,23,24,25]. 

The publication dates ranged from 2018−2021, and where reported, the data included ranged from January 2010 to April 2019 (Table 1). Study designs varied widely, including retrospective cohort [15,20,22,23], case-control [18,19,21], pre/post (or before/after) interventions [24,25,26], cross-sectionals studies [15], combination designs [16], and randomized controlled trials [25]. Observational studies overall were of moderate quality while the single RCT in this review had a high risk of bias (Appendix A). Data sources included medical records, electronic medical records (EMR), or standardized forms. 

Settings of care varied, though all studies were performed with hospitalized inpatients. Six papers included patients treated in the intensive care unit (ICU) as well as general inpatient populations [18,19,20,22,23,27], two papers included patients treated in the neonate and pediatric intensive care unit (N/PICU) [15,21], and one paper included patients treated in the emergency department (ED) [14]. 

Five papers [16,21,22,25,27] reported exclusively on pediatric patients (i.e., under 18 or 21 years old), seven [14,17,18,19,22,23,27] reported on adults or patients of mixed ages, and one paper [16] did not report the age of their included patients. With the exception of five papers [15,17,18,25,26], the majority reported peripheral and/or cerebrospinal white blood cell (WBC) counts and CSF laboratory tests. Three papers [16,19,23] reported proportions of immunosuppressed patients. All but two papers [14,16] reported the distribution of the pathogens detected in each study group.

### 3.2. Length of Hospital Stay

Across eleven studies, a statistically significant reduction in the average length of hospital stay (MD [95% CI]: −1.20 days [−1.96, −0.44]) was found among patients tested with the multiplex ME panel compared with standard of care (Figure 2). 

The stratification by age showed that among the four studies [19,20,21,24,25] which included pediatric patients exclusively, there was a nonsignificant reduction in the average length of hospital stay (MD [95% CI]: −1.09 [−2.23, 0.05]; Appendix A). Among the seven studies including mixed/adult populations [14,17,18,19,22,23,27], there was a significant association between the multiplex ME panel and reduced length of hospital stay (MD [95% CI]: −1.33 [−2.40, −0.26]).

### 3.3. Length of Acyclovir Therapy

Seven studies [14,18,19,20,21,23,25] reported the average length of acyclovir therapy between patients tested with the multiplex ME panel and control groups. There was a statistically significant reduction in the length of acyclovir therapy (MD [95% CI]: −1.14 days [−1.78, −0.50]; Figure 3).

Among the three papers [19,20,21,25] that exclusively included pediatric patients, a significant reduction was also observed (MD [95% CI]: −1.73 [−2.59, −0.86]; Appendix A). A non-significant reduction was observed in the remaining four papers [15,19,20,24] (MD [95% CI]: −0.43 [−1.60, 0.73]).

### 3.4. Days with Antibiotic Therapy

Six studies [14,18,19,20,21,24] compared the length of treatment with antibiotics between patients tested using the multiplex ME panel and control groups. There was a nonsignificant reduction in the average number of days with antibiotics (MD [95% CI]: −1.01 [−2.39, 0.37]; Figure 4). 

In the three papers [21,22,25] that included pediatric samples, a significant reduction in the number of days with antibiotics was observed (MD [95% CI]: −1.85 [95% CI: −2.50, −1.21]; Appendix A). In the remaining three papers [15,19,20], the meta-analysis found no association (MD [95% CI]: 0.18 [−1.39, 1.76]).

## 4. Discussion

These results show that the use of the multiplex ME panel in clinical practice is associated with a significant reduction in the average length of stay and average length of acyclovir therapy. While a reduction of 1.01 days was observed for the number of days with antibiotic therapy, this reduction was not statistically significant. The stratification of the outcomes among studies reporting on pediatric populations exclusively showed broadly similar results for length of stay, and suggested a greater impact for length of acyclovir therapy and duration of antibiotic therapy. Based on the rapid turnaround and diagnostic yield of the panel, the results of this meta-analysis support that the panel may streamline patient management (targeting therapy, discontinuing unnecessary therapy, avoiding additional testing and imaging, etc.). 

These results are broadly similar to a recently published technical note [8] that examined the impact of the multiplex ME panel in ten controlled studies (eight of which overlap with the current systematic review). Of these ten studies, eight found statistically significant reductions in either length of acyclovir or antibiotic therapy (four of eight for antibiotic duration), and two out of eight studies found a significant reduction of >24 h in length of stay. The current review builds on these signals by providing a quantitative indicator of the extent to which this reduction may be observed in a pool of studies. 

Prediction models intended to assess the likelihood of bacterial meningitis in patients with suspected CNS infection have historically performed poorly [26], suggesting rapid diagnostic testing is a potential avenue to facilitate faster and confirmed diagnosis. The results from this meta-analysis provide quantitative evidence for a dual economic and clinical impact of the multiplex ME panel. It has been reported that the multiplex ME panel considerably decreased the turnaround time for pathogen identification, from over 24 h by standard approaches to a few hours [8]. The rapidity of diagnosis likely contributes to an overall reduced hospital stay, as identifying the suspected pathogen may decrease the time to targeted therapy. 

It is worth noting that the use of length of stay as a proxy for clinical utility also needs to consider the accuracy of the results. Untimely or erroneous test results may lead to early mortality, thereby reducing length of stay and falsely indicating an improvement in the quality of care. Therefore, proper contextualization should be provided in studies using this outcome to assess the utility of the multiplex ME panel. 

The Mina 2019 study featured an apparently greater reduction in hospital length of stay than other studies included for this outcome. This study required a diagnosis of bacterial meningitis among all participating patients, which may explain why the reduction was more pronounced [22]. A sensitivity analysis was performed whereby this study was excluded and the meta-analysis was performed among the remaining 10 studies. Results were largely unchanged (MD [95% CI]: −1.05 days [−1.74, −0.36]), suggesting that this study alone had a minimal impact on the findings. Given mean cost per day of meningitis cases has been reported at $756 per patient per day in a prior study of aseptic meningitis patients, the observed one day decrease in this meta-analysis may have significant financial implications [28]. 

It has been reported in other work involving the multiplex ME panel that changes to empiric therapy are made after positive results are received in the majority of cases [29,30]. Furthermore, in a study conducted in Colombia, the introduction of the multiplex ME panel decreased time to targeted therapy from 195 to 2.1 h [18]. This meta-analysis provides additional context to the clinical endpoints from smaller studies by quantifying the extent to which acyclovir therapy was significantly reduced. The −1.14 day decrease in acyclovir observed in this meta-analysis may have important implications on acute kidney injury (AKI) as AKI has been reported in various studies to occur as soon as one to three days after initiation of therapy [31,32,33,34]. In the stratification for pediatric studies, while the number of included studies was small, it was noted that there appeared to be an increased impact of the multiplex ME panel as the point estimates of both antibiotic days and acyclovir length were lower than that of the main analysis. The identity of causative pathogens may present particular importance to this patient population, and merits further investigation.

A statistically significant increase in the number of days of antibiotics was observed across two out of the three studies that included adult patients, which was surprising [18,19]. After weighting, the association was primarily driven by a study by Evans et al. (2020) [19]. The authors mentioned that the prolonged antibacterial therapy observed across patients in the Panel group could be explained by a knowledge gap by the providers following the implementation of the Panel. These outcomes could possibly be improved by adding the input of an antimicrobial stewardship team in future studies. Furthermore, while the data transformation of medians and inter-quartile ranges into means and standard deviations resulted in a statistically significant reduction in the number of days of antibiotics, it is worth noting that the original publication reported a nonsignificant difference in the number of days (P-value: 0.28). This highlights the need to interpret these results with caution, as stratified meta-analyses with a low number of studies (in this case, three) are subject to misinterpretation due to the limited contributing data.

### Limitations

These results should be interpreted in light of certain limitations. First, in order to standardize outcomes reported from publications into a format suitable for pooling and analysis, data transformations were required (the majority of studies reported medians and inter-quartile ranges for their outcomes, not means and standard deviations). Such transformations required assumptions about the distribution of the data [11].

Second, the overall number of publications retained was relatively small, particularly for the stratified analysis of the length of treatment with antibiotic and acyclovir. As mentioned in the Discussion, this may have resulted in a single paper having had a material impact on the results. This also makes the findings of this review prone to publication bias, and caution is required when synthesizing such a small number of studies.

Third, the literature search did not retrieve many large-scale randomized studies. Only one of the retained studies was a randomized controlled trial, and the sample size of the other studies was relatively modest. Furthermore, many studies used historical controls, and it is unclear whether these studies could properly account for changes in the outcomes that were unrelated to the intervention (e.g., decreasing length of stay due to improving standards of care over time). Quality assessment tools suggested moderate quality among observational studies and some concerns of bias from the single RCT. Together, this indicates that there is a need for more large-scale high-quality studies to help bolster the level of evidence for the potential benefit of the multiplex ME panel.

Fourth, as the purpose of this review was to evaluate the impact of the multiplex ME panel within clinical contexts “at large”, results were not stratified by pathogen. It would be valuable to assess whether these outcomes differ between studies based on the prevalence of each pathogen among studies retained. For example, guidelines for the diagnosis of herpes simplex virus (HSV) encephalitis recommend to repeat a PCR test among patients with an initial negative PCR result and suspected features of HSV encephalitis (albeit this is seldom carried out in clinical practice) [35,36]. Consequently, results among patients with suspected HSV encephalitis are likely to be different than for patients with other suspected pathogens.

Lastly, there is a paucity of field data in how the Panel was used in each study, as well as the heterogeneous practices of each hospital under standard of care, which could impact outcomes such as the length of stay. This limitation has been noted in other research involving the multiplex ME panel [8]. There was no information on whether the clinical flow was the same or how antimicrobial stewardship is managed for each setting. In order to further evaluate how a hospital’s guidelines, culture, and resources may affect these results, future studies may consider alternative field-based study designs such as time-and-motion studies, which can help assess the amounts of time spent in each stage of identifying and treating the causal pathogen in a given care setting.

## 5. Conclusions

These findings suggest that the use of the multiplex ME panel is associated with a significant reduction in the hospital length of stay and length of acyclovir therapy, as well as a potential reduction in the number of days with antibiotic therapy. The multiplex ME panel has the potential to be an important component of antibiotic stewardship programs, and its clinical benefits may translate into more effective and targeted patient management. Further research may include large-scale randomized trials to further bolster the validity of these findings.

## Figures and Tables

**Figure 1 antibiotics-11-01028-f001:**
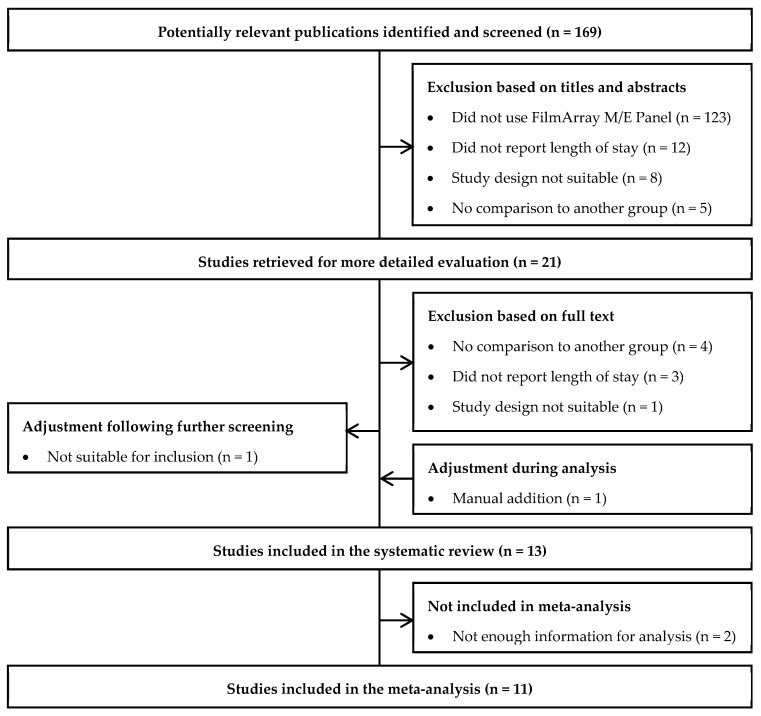
Study Selection Flow Diagram.

**Figure 2 antibiotics-11-01028-f002:**
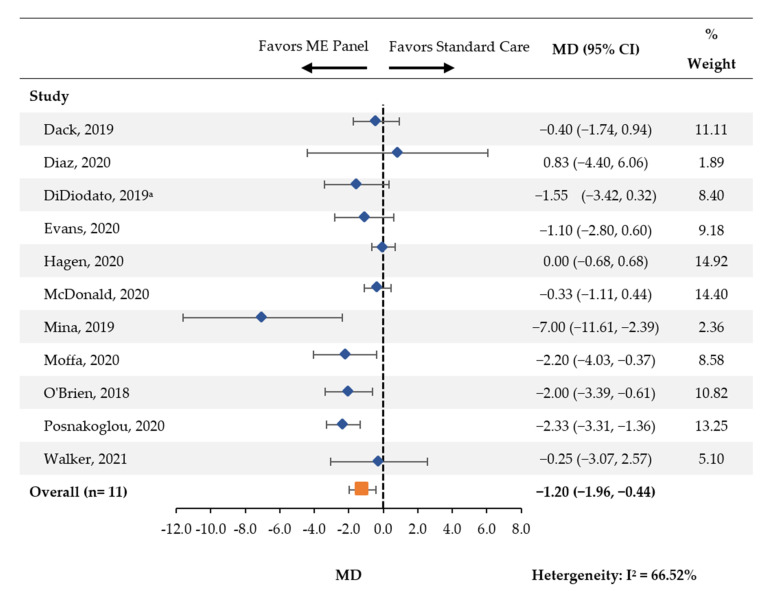
Hospital Length of Stay. CI: confidence interval; MD: mean difference; ME: meningitis and/or encephalitis. ^a^ Analysis was performed on the subgroup of patients whose time to discharge was ≤18 days, n = 95.

**Figure 3 antibiotics-11-01028-f003:**
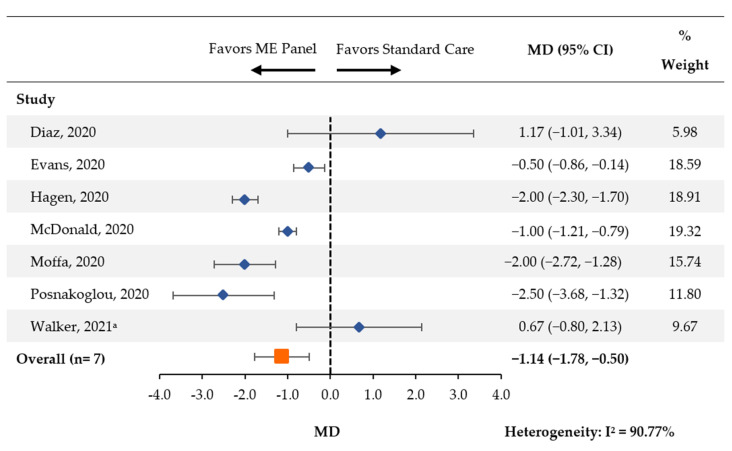
Length of Acyclovir Treatment. CI: confidence interval; MD: mean difference; ME: meningitis and/or encephalitis. ^a^ One patient (of 19) in the pre-intervention group received an antiviral that was not acyclovir.

**Figure 4 antibiotics-11-01028-f004:**
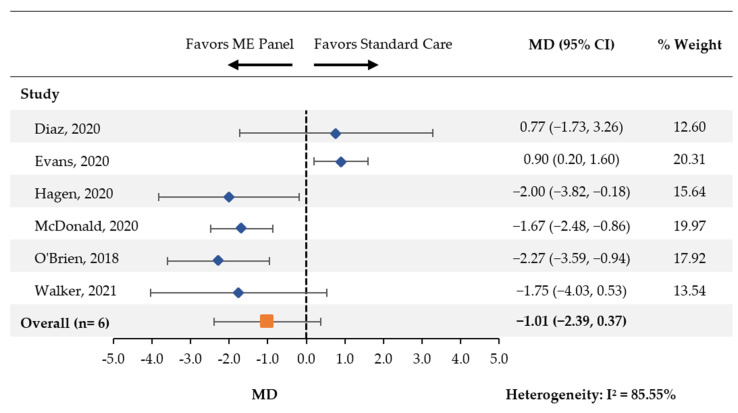
Days of Treatment with Antibiotics. CI: confidence interval; MD: mean difference; ME: meningitis and/or encephalitis.

**Table 1 antibiotics-11-01028-t001:** Characteristics of Studies Included in the Systematic Review.

First Author, Year	Population	Reported Control Testing	Month[s] and Year[s] Captured	ME Group		Control Group	
				N	Age, mean ± SD * (Years)	N	Age, mean ± SD * (Years)
Dack K, 2019	Adult Patients	CSF culture and GS	7/8/2015−8/6/2017	47	−	50	−
Diaz KMO, 2020	Adult Patients	CSF culture and GS, CSF fungal culture, blood cultures, CT and MRI imaging (suspected encephalitis cases), India ink stain, CMV PCR and CrAg (patients with HIV diagnosis).	Before and after implementation of panel (May 2016)	46	43.75 ± 5.25	52	35.75 ± 5
DiDiodato G, 2019	Unspecified	CSF cell count, CSF glucose and protein, CSF culture and GS, CSF fungal culture and stains, HSV PCR (send-out), EV PCR (send-out)	4/1/2016−31/3/2018	53	43.99 ± 25.7	64	51.3 ± 20.7
Evans M, 2020	Adult and Pediatric Patients	CSF cell count, CSF glucose and protein, CSF bacterial/fungal culture, HSV PCR (batched testing), EV PCR (in-house), CMV PCR (send-out), HHV-6 PCR (send-out), EBV PCR (send-out)	04/01/2016−12/01/2017	76	Min age: 0 Max age: 89	132	Min age: 0 Max age: 89
Hagen A, 2020	Pediatric Patients	Viral PCR send-out tests (HSV-1/2, EV, and HHV-6)	01/2012−02/2017	46	0.8 ± 1.4	46	0.7 ± 1.3
McDonald D, 2020	Pediatric Patients	CSF cell count, culture, molecular respiratory pathogen panel	01/2015−09/2018	61	1.3 ± 1.0	186	1.2 ± 0.9
Mina Y, 2019	Unspecified	CSF cell count, CSF culture, blood culture	01/2010−06/2018	8	40 ± 26	23	43 ± 20
Moffa MA, 2020	Adult Patients	CSF cell count, CSF glucose and protein, CSF culture, HSV PCR (send-out), VSV PCR (send-out), CMV PCR (send-out)	10/2016−9/2018	79	49.9 ± 17.5	81	50.6 ± 20.1
Mostyn A, 2020	Unspecified	CSF cell count, CSF culture and GS, latex agglutination tests (*N. meningitidis* A, B, C, Y, and W135; *E. coli* K1; *H. influenzae* Type B; *S. pneumoniae*; *S. agalactiae*), bacterial PCR send-out tests (*N. meningitidis*, *E. coli* K1, *H. influenzae*, *S. pneumoniae*, *S. agalactiae*), viral PCR send out tests (HSV 1, HSV 2, VZV, HHV-6), CrAg	12/2016−07/2017	16	−	18	−
Nabower AM, 2019	Pediatric Patients	CSF cell count, CSF culture, EV PCR, HSV PCR	6/2015−7/2017	223	<30 days: 67 (30.0%) 30−90 days: 100 (44.8%) >90 days: 57 (25.6%)	348	<30 days: 121 (34.8%) 30−90 days: 129 (37.1%) >90 days: 98 (28.2%)
O’Brien MP, 2018	Pediatric Patients	CSF cell count, CSF glucose and protein, CSF culture, viral PCR on-site tests (HSV and VZV), viral PCR send-out tests (HPeV and EV)	11/2014−5/2017	29	−	36	−
Posnakoglou L, 2020	Pediatric Patients	CSF cell count, CSF glucose and protein, CSF culture and GS, viral PCR send-out tests (not defined, ordered at physician discretion)	4/2018−4/2019	71	2.1 ± 4.4	71	1.1 ± 2.2
Walker M, 2021	Adult Patients	CSF cell count, CSF diagnostics (not defined)	6/2015−9/2016	91	−	72	−

CMV: cytomegalovirus; CrAg: cryptococcal antigen; CSF: cerebrospinal fluid; CT: computed tomography; EBV: Epstein-Barr virus; EV: enterovirus; GS: Gram stain; HHV-6: human herpesvirus 6; HIV: human immunodeficiency virus; HPeV: human parechovirus; HSV: herpes simplex virus; ME: meningitis and/or encephalitis; MRI: magnetic resonance imaging; PCR: polymerase chain reaction; SD: standard deviation; VSV: vesicular stomatitis virus; VZV: varicella zoster virus. * Unless otherwise indicated.

## Data Availability

Not applicable.

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
