# Peer review of "Assessment of the Impact of a Meningitis/Encephalitis Panel on Hospital Length of Stay: A Systematic Review and Meta-Analysis"

_antibiotics, 2022, doi:10.3390/antibiotics11081028_

Round 1

Reviewer 1 Report

In "Assessment of the Impact of a Meningitis/Encephalitis Panel on Hospital Length of Stay: A Systematic Review and Meta-Analysis", the authors did a meta-analysis on the literature on multiplex ME panel and the length of hospital stay, acyclovir and antibiotic therapy.  The manuscript is well constructed. I would like to point the following revisions:

1. The authors affiliations are not properly assigned. Please correct. 

2. The conflict of interests should be clearly indicated as some authors are employees of BioFire Diagnostics.

3. Please indicate the antibiotics used in the studies. Is there any correlation with the antibiotic type and the LOS or acyclovir therapy?

4. Please explain in the discussion section how the BioFire FilmArray will improve the therapy. 

Reviewer 2 Report

This study was conducted by BioFire Diagnostics with the help of a consulting company (Analysis Group, Inc) that has received funding from BioFire Diagnostics to publicize on the BioFire FilmArray Meningitis/Encephalitis Panel. The sponsor was involved in the study design, data collection, data analysis, manuscript preparation, and publication decisions. Therefore, I do not think this work meets the minimal ethical criteria for publication in a scientific journal.

Reviewer 3 Report

The manuscript seems to be well-written. I have some comments.

1.         Figures are a little bit blurred. How about increasing the resolution?

2.         (Table 1) I think that you should add what panel or method was used in the control group for each previous study. Meaning of the “reduction of 1 day” varies depending on the control group.

3.         (Discussion) Although it was found that multiplex ME panel was associated with a reduction of 1 day for some outcomes, what is the clinical significance for it? Is the reduction of 1 day clinically meaningful?

Round 2

Reviewer 2 Report

Dear Authors,
I acknowledge receipt of your reply letter. As a independent reviewer, my opinion is that your work is not scientifically-independant and therefore does not meet the ethical criteria for publication in a scientific, peer-reviewed journal. To me, your work corresponds more to a document intended for regulatory agencies such as FDAThe decision and responsibility to publish your work is up to the Editor.
Sincerely